# Etiology, Pathogenesis, Diagnosis, and Practical Implications of Hepatocellular Neoplasms

**DOI:** 10.3390/cancers14153670

**Published:** 2022-07-28

**Authors:** Prodromos Hytiroglou, Paulette Bioulac-Sage, Neil D. Theise, Christine Sempoux

**Affiliations:** 1Department of Pathology, Aristotle University School of Medicine, 54124 Thessaloniki, Greece; 2INSERM, BRIC, U1312, University Bordeaux, F-33000 Bordeaux, France; paulette.bioulac@u-bordeaux.fr; 3Department of Pathology, New York University Grossman School of Medicine, New York, NY 10016, USA; neil.theise@nyulangone.org; 4Service of Clinical Pathology, Institute of Pathology, Lausanne University Hospital, University of Lausanne, CH-1007 Lausanne, Switzerland; christine.sempoux@chuv.ch

**Keywords:** hepatocellular adenoma, hepatocellular carcinoma, molecular pathology, histologic diagnosis, diagnostic algorithm, LI-RADS, frequently asked questions

## Abstract

**Simple Summary:**

In recent years, significant progress has been made in elucidating the mechanisms via which hepatocellular neoplasms, i.e., hepatocellular adenoma and hepatocellular carcinoma, arise. Hepatocellular carcinoma usually occurs in livers with chronic disease, due to deregulation of important intracellular pathways of signal transmission. Recent studies suggest that subclassification of hepatocellular carcinoma is practically useful. On the other hand, subclassification of hepatocellular adenomas has been well established through correlation of molecular alterations with morphology and protein expression. Advances in hepatic imaging have resulted in a new approach for diagnostic assessment of lesions arising in advanced chronic liver disease. Histologic examination, aided by immunohistochemistry, is the gold standard for the diagnosis and subclassification of hepatocellular neoplasms, while clinicopathologic correlation is essential for best patient management. We summarize the etiology and pathogenesis of hepatocellular neoplasms, provide practical information for their histologic diagnosis, and address various frequently asked questions regarding their diagnosis and practical implications.

**Abstract:**

Hepatocellular carcinoma (HCC), a major global contributor of cancer death, usually arises in a background of chronic liver disease, as a result of molecular changes that deregulate important signal transduction pathways. Recent studies have shown that certain molecular changes of hepatocarcinogenesis are associated with clinicopathologic features and prognosis, suggesting that subclassification of HCC is practically useful. On the other hand, subclassification of hepatocellular adenomas (HCAs), a heterogenous group of neoplasms, has been well established on the basis of genotype–phenotype correlations. Histologic examination, aided by immunohistochemistry, is the gold standard for the diagnosis and subclassification of HCA and HCC, while clinicopathologic correlation is essential for best patient management. Advances in clinico-radio-pathologic correlation have introduced a new approach for the diagnostic assessment of lesions arising in advanced chronic liver disease by imaging (LI-RADS). The rapid expansion of knowledge concerning the molecular pathogenesis of HCC is now starting to produce new therapeutic approaches through precision oncology. This review summarizes the etiology and pathogenesis of HCA and HCC, provides practical information for their histologic diagnosis (including an algorithmic approach), and addresses a variety of frequently asked questions regarding the diagnosis and practical implications of these neoplasms.

## 1. Introduction

In the past decade, application of novel methodologies of molecular medicine in hepatocellular neoplasms has significantly improved our understanding of the pathogenesis of hepatocellular adenoma (HCA) and hepatocellular carcinoma (HCC), as well as provided useful new markers for pathologic diagnosis. At the same time, advances in clinico-radio-pathologic correlation have resulted in a new approach for the diagnostic assessment of focal hepatic lesions arising in advanced stage chronic liver disease by imaging, termed LI-RADS. In addition to providing new diagnostic and prognostic markers, elucidation of the molecular pathways of these neoplasms also has significant implications for treatment. This is particularly important for patients with HCC, for whom precision oncology strategies are finally starting to emerge, following many years of intensive research. This article briefly reviews the etiology and pathogenesis of HCA and HCC, provides practical information for their histologic diagnosis, and addresses a variety of frequently asked questions regarding the diagnosis and practical implications of these neoplasms.

## 2. Etiology and Pathogenesis of Hepatocellular Adenomas

It is now well recognized that hepatocellular adenoma (HCA), occurring mainly in young women taking oral contraception (OC), is a heterogeneous entity comprising different morpho-molecular subtypes, with various clinical and etiological backgrounds, risk for complications (bleeding and malignant transformation), and pathogenesis [1]. While most HCAs appear in normal liver, several clinical conditions and genetic syndromes have also been found to be linked to the development of HCAs [1].

The first well-recognized subtype is related to *HNF1A*-inactivating mutations (H-HCA). These tumors may be solitary or multiple, or they may occur in the context of liver adenomatosis. H-HCA is usually characterized by steatosis within the lesion and has a low risk of complications.

The second subtype is the inflammatory hepatocellular adenoma (IHCA), often developing on a background of NAFLD or in the context of alcohol consumption, predominantly but not exclusively in obese women. These lesions are often multiple. Typically characterized by sinusoidal dilatation and inflammation, IHCAs are related to different mutations leading to IL6/JAK/STAT inflammatory pathway activation.

A third subtype is the HCA with β-catenin-activating mutations (b-HCA). A proportion of these mutations occur in IHCA, thus giving rise to b-IHCA. By contrast with the other subtypes, b-(I)HCAs are overrepresented in men and have a higher risk of malignant transformation. This risk depends on the level of activation of the β-catenin pathway, which is linked to the type of *CTNNB1* mutation that results also in different immunohistochemical features [2].

A recently identified fourth HCA subtype is related to activation of the sonic hedgehog pathway (shHCA). These tumors are prone to bleeding, even when small, and can be recognized by argininosuccinate synthase 1 (ASS1) overexpression on immunohistochemistry [3,4]. This subtype has been described so far only in women, often overweight, and in the context of the metabolic syndrome.

Figure 1 illustrates the different subtypes of HCA with their principal immunohistochemical characteristics: H-HCA and liver fatty-acid-binding protein (LFABP), IHCA and C-reactive protein (CRP), b-HCA/b-IHCA and glutamine synthetase (GS), and shHCA and ASS1.

## 3. Diagnosis and Subtyping of Hepatocellular Adenomas

The histologic diagnosis of HCA requires careful assessment of representative hematoxylin and eosin (H&E)-stained sections. HCAs are characterized by a benign hepatocellular proliferation, devoid of portal tracts. “Unpaired” arteries (i.e., arteries unaccompanied by veins or bile ducts) are present among the neoplastic cells. Other characteristic features include steatosis, inflammation, sinusoidal dilatation, and/or areas of hemorrhage. After H&E assessment, immunohistochemical evaluation follows with specific antibodies recognizing the targets identified by the genotype–phenotype studies [1,2]. An algorithm for the diagnosis is proposed in Figure 2.

## 4. Etiology and Pathogenesis of Hepatocellular Carcinoma

Hepatocellular carcinoma (HCC) usually arises in livers with chronic disease, and it is most often discovered when disease has reached an advanced stage, traditionally known as cirrhosis. The most common chronic diseases that are associated with HCC are chronic hepatitis B, chronic hepatitis C, and alcoholic liver disease, accounting together for 84% of the cases occurring globally in 2015 [5]. In the meanwhile, nonalcoholic steatohepatitis (NASH), associated with the metabolic syndrome, is emerging as a major risk factor for HCC [6]. Other risk factors include hereditary metabolic disorders (such as hemochromatosis, α1-antitrypsin deficiency, and tyrosinemia), aflatoxin B1 exposure (in individuals chronically infected with HBV), and tobacco smoking. Chronic liver diseases other than those mentioned above (e.g., autoimmune hepatitis, primary biliary cholangitis, primary sclerosing cholangitis, and Wilson disease) are uncommonly associated with development of HCC.

HCC arising in noncirrhotic livers is often caused by HBV, which is a virus with known carcinogenic effects. HBV DNA insertion in the host genome can deregulate genes involved in cell signaling and replication (such as *TERT, PDGFR, MLL4,* and *CCNE1*), while the HBV X protein transactivates genes involved in signal transduction pathways and inhibits *TP53* expression [7,8,9]. NASH and hereditary hemochromatosis are also increasingly recognized as causes of HCC arising in noncirrhotic livers [6,10]. However, HCC can also arise in apparently normal liver. Some of these cases may represent evolution of HCA (mostly b-HCA and b-IHCA) to HCC (discussed in the previous sections), while others, usually occurring in older individuals, remain unexplained. A special HCC subtype arising in normal livers of young individuals is fibrolamellar carcinoma, which is associated with a characteristic somatic gene fusion, *DNAJB1–PRKACA*, resulting from deletions in chromosome 19 and activating protein kinase A [11].

In chronic liver diseases, continuous cell loss results in cell proliferation occurring in a noxious microenvironment, characterized by oxidative stress due to chronic inflammation, overexpression of growth factors, and epigenetic changes due to derangements of DNA methyltransferases [12,13,14]. Thus, the possibility of mutations that initiate or promote carcinogenesis is increased, while mutations providing survival benefits to hepatocytes favor clonal expansion. This process is accelerated in the advanced stages of chronic liver diseases when vascular changes, including intrahepatic vein thrombosis and vascular reorganization, result in extensive cell loss. In that setting, hepatic regeneration largely depends on progenitor cell proliferation due to senescence of hepatocytes. Therefore, critical mutations in progenitor cells have the potential to produce large numbers of clonally expanding hepatocytes with increased likelihood to progress to precancerous lesions and then to HCC.

The diverse molecular changes that are associated with HCC have been recently reviewed [15]. Whole-exome and whole-genome sequencing studies have revealed 40–60 somatic coding mutations per HCC, including 4–6 driver mutations [16]. The most frequent mutations in HCC are those involving the promoter of telomerase reverse transcriptase (*TERT*), occurring in 60% of cases [17]. In an additional 30% of HCCs, *TERT* is deregulated by other molecular mechanisms, such as viral insertion [18]. *TERT* promoter mutations have also been detected in precancerous nodules and are considered an early event in hepatocarcinogenesis [19]. Other frequently mutated genes in HCC include *CTNNB1, TP53, RB1, ARID1A, ARID2, AXIN1*, albumin, and apolipoprotein B [20,21,22]. The mutations occurring in hepatocarcinogenesis can disrupt various signal transduction pathways, such as telomere maintenance (*TERT*), cell-cycle control (*TP53, CDKN2A*), Wnt/β-catenin (*CTNNB1, AXIN1*), epigenetic (*ARID1A, ARID2, MLL2*), and oxidative stress (*NFE2L2, KEAP1*) [17,23,24]. “Druggable” genetic alterations are under intense investigation because, at the present time, targeted therapeutic agents for HCC are limited to a small number of multikinase inhibitors. On the other hand, understanding the interaction between neoplastic cells and their microenvironment will be crucial for identifying biomarkers and developing new therapies based on immune checkpoint inhibition [25]

Recent studies have shown that certain molecular changes in HCC are associated with specific clinicopathologic features and prognosis, suggesting the possibility of a molecular classification for the future [26,27,28,29]. This active research has resulted in the recognition of several HCC subtypes (also called “variants”) that hold promise for a more personalized treatment of HCC patients. Eight HCC subtypes, considered to represent distinct clinicopathological/molecular entities and accounting together for up to 35% of HCCs, have been included in the latest edition of the WHO classification of liver tumors [30]. The characteristic features of these subtypes are briefly presented in Section 5. It should be kept in mind that subclassification of HCC is a work in progress that will achieve significantly more importance if it becomes useful from a therapeutic point of view.

## 5. Diagnosis of Hepatocellular Carcinoma

Diagnosis of HCC is traditionally made by histologic examination of biopsy, surgical, or autopsy specimens, and it is based on the recognition of two basic attributes in the histologic material: (i) hepatocellular differentiation, and (ii) malignancy. Features suggesting hepatocellular differentiation include resemblance of neoplastic cells to hepatocytes, bile production by neoplastic cells, positive immunostaining of neoplastic cells for “hepatocytic” markers, such as arginase-1 and carbamoyl phosphate synthetase-1 (recognized by the antibody HepPar1), and detection of albumin mRNA by in situ hybridization. Except for bile production by neoplastic cells, none of the other features mentioned above is entirely specific for HCC. On the other hand, features indicating malignancy include stromal invasion, vascular invasion, metastatic spread, trabeculae thicker than three cells, and immunopositivity of neoplastic cells for oncofetal antigens α-fetoprotein and/or glypican-3.

In addition to the most common trabecular growth pattern, HCCs often display solid (compact), pseudoglandular, and macrotrabecular patterns of growth, including combinations thereof. Similar to hepatocytes, the neoplastic cells may contain fat, glycogen (resulting in clear cell change), hyaline bodies, Mallory–Denk bodies, or pale bodies. Scattered arteries unaccompanied by veins or bile ducts (i.e., “unpaired” arteries) are a characteristic histologic finding. Portal tracts are not a feature of classic HCC, except in the invasive front of some tumors. Similar to other carcinomas, HCC is also histologically classified as well, moderately and poorly differentiated [30] (Figure 3). Histologic diagnosis of poorly differentiated HCC is often difficult and requires immunohistochemical stains in support of the diagnosis (arginase-1, HepPar1, α-fetoprotein, and glypican-3), as well as appropriate markers for other tumors that are included in the differential diagnosis, on a case-per-case basis.

On the other hand, some HCCs are difficult to recognize histologically, especially in biopsy material, because of well-differentiated features. Absence of portal tracts and presence of unpaired arteries in the biopsy material are features suggesting hepatocellular neoplasm, but do not allow distinction between HCA and well-differentiated HCC, while thin cell plates (<3 cells) do not exclude HCC. This difficult differential diagnosis is discussed below (see FAQ 1). It is emphasized that correlation of clinical, radiologic, and pathologic findings is essential for correct classification of difficult cases. This is particularly true in the interpretation of biopsy material from small (<2 cm) nodular lesions in cirrhotic livers, where the differential diagnosis includes large regenerative nodule, dysplastic nodule (low or high grade), early HCC, and classic HCC (see Section 6). This interpretation is facilitated when biopsy material from the hepatic parenchyma away from the lesion is available for comparison.

Early HCC (eHCC) has recently been recognized as a distinct step in hepatocarcinogenesis, characterized by ability for stromal invasion, but not for vascular invasion or metastatic spread [31]. By definition, eHCC is a well-differentiated, early-stage tumor that measures less than 2 cm in diameter. On gross examination, eHCC often appears vaguely nodular, without distinct pushing boundaries or pseudocapsule, whereas small HCC of the classic (also called “progressed”) type usually has distinct boundaries marked by a pseudocapsule comprising compressed portal tracts or disease-associated scars [32]. Small classic HCCs tend to be better differentiated than larger ones, but have similar histologic features. On the other hand, many histologic features of eHCCs are reminiscent of those seen in high-grade dysplastic nodules. Early HCCs are usually composed of crowded, relatively small neoplastic cells, arranged in thin trabeculae and occasional small pseudoglandular structures. High cellularity (more than twice that of the surrounding parenchyma) and indistinct borders are characteristic features on low-power microscopic examination. Unpaired arteries are usually sparse and small, as compared to those of classic HCC. “Entrapped” portal tracts may be present in eHCC, especially in peripheral regions of the lesion. Steatosis is also often seen in eHCC, and it has been attributed to reduced oxygen supply compared to surrounding parenchyma [32]. On occasion, histologic examination of hepatic nodules may reveal classic HCC arising within eHCC (Figure 4). The vascular supply of eHCC (portal tract vessels and poorly developed unpaired arteries) significantly overlaps with that of dysplastic nodules; therefore, distinction between these lesions with imaging methods is difficult to impossible. The histologic features distinguishing eHCC from high-grade dysplastic nodules are discussed below (see Section 6).

Table 1 provides a comparison of the etiology, pathogenesis, and diagnostically useful histopathologic features of HCA and HCC.

### HCC Subtypes

The *steatohepatitic subtype* of HCC occurs usually, but not exclusively, in patients with metabolic syndrome or alcohol use and is characterized by histologic features similar to those of steatohepatitis occurring in nontumorous liver, i.e., macrovesicular steatosis, inflammation, ballooned cells, Mallory–Denk bodies, and pericellular fibrosis [33,34] (Figure 5a). This subtype was found to be associated with frequent IL6/JAK/STAT pathway activation, without *CTNNB1, TERT*, and *TP53* alterations [29]. At this point in time, steatohepatitic HCC does not seem to prognostically differ from average classic HCC.

The *clear cell subtype* owes its appearance to glycogen accumulation in tumor cells, thus simulating clear-cell carcinoma of the kidney and other organs (Figure 5b). No characteristic molecular alterations have been found in this subtype, which appears to be associated with a better-than-average prognosis [35]. Distinction from metastatic renal cell carcinoma may require immunohistochemical stains for hepatocytic markers (arginase-1, HepPar1) and renal transcription factor PAX-8.

The *macrotrabecular massive* subtype is histologically characterized by thick trabeculae, although the exact thickness (>6 cells vs. ≥10 cells thick) differs among authors [36] (Figure 5c). This subtype is associated with high serum α-fetoprotein and poor prognosis [29]. *TP53* mutations and *FGF19* amplifications are common in these tumors.

The *scirrhous subtype* is characterized by diffuse fibrosis, and it has been associated with *TSC1/TSC2* mutations [29] (Figure 5d). The prognosis of this subtype does not appear to differ from the average classic HCC. On histologic examination, this subtype should be distinguished from cholangiocarcinoma. Immunohistochemistry for hepatocytic markers arginase-1 and HepaPar1 is useful in this regard, whereas cytokeratin 7 is positive in most scirrhous HCCs and almost all cholangiocarcinomas.

The *chromophobe*
*subtype* is characterized by light staining cytoplasm of the neoplastic cells, mostly bland nuclei, as well as scattered cells with large atypical nuclei. Another characteristic feature is the presence of scattered cystic spaces, filled with serum-like material. On a molecular basis, this subtype is characterized by alternative lengthening of telomeres, a mechanism for telomere preservation without *TERT* promoter mutation [37]. The prognosis of this subtype does not appear to differ from the average classic HCC.

The *fibrolamellar subtype* has long been considered a distinctive HCC variant occurring in young individuals (median age: 25 years) without liver disease. These tumors are well differentiated and consist of groups and trabeculae of large polygonal cells, separated by bands of lamellar fibrosis. The neoplastic cells have abundant eosinophilic cytoplasm, often displaying pale bodies, as well as large nuclei with prominent nucleoli (Figure 5e). In contrast to most other HCCs, those of the fibrolamellar subtype are positive for cytokeratin 7 and CD68. Almost all fibrolamellar HCCs have the characteristic somatic gene fusion *DNAJB1–PRKACA*, the detection of which can aid diagnosis [11]. The prognosis of fibrolamellar HCC is similar to that of classic well-differentiated HCC occurring in noncirrhotic liver.

The *neutrophil-rich subtype* is characterized by abundant intratumoral neutrophils, due to granulocyte colony-stimulating factor (G-CSF) produced by neoplastic cells. Most tumors are poorly differentiated and may have sarcomatoid areas. The patients have elevated peripheral white blood cell counts, serum IL-6 levels, and often serum C-reactive protein. The prognosis of this subtype is worse than the average classic HCC [30].

The *lymphocyte-rich subtype* is characterized by abundant intratumoral lymphocytes. Cases tested for Epstein–Barr virus (EBV) were found to be negative. No prognostic significance has been attributed to this subtype. The lymphocyte-rich subtype should be distinguished from *lymphoepithelioma-like HCC*, a rare, poorly differentiated carcinoma, composed of tumor cells growing in poorly defined groups within a dense lymphoplasmacytic infiltrate [36,38]. Most cases of this neoplasm, which has similar histologic features to nasopharyngeal carcinoma and lymphoepithelioma-like carcinomas arising in other organs, have also been found to be negative for EBV.

In addition to lymphoepithelioma-like HCC, *sarcomatoid HCC* is another poorly differentiated variant that has not been recognized as a separate subtype in the latest edition of the WHO classification of liver tumors [30]. However, sarcomatoid HCC merits specific mention because it has a poor prognosis, as well as a spindle cell morphology that mimics various sarcomas [39] (Figure 5f). Extensive sampling may be required to reveal areas of typical HCC in these tumors, while immunohistochemical stains demonstrating expression of epithelial and hepatocytic markers can be useful, especially in cases with limited histologic material. Heterologous differentiation may be found in these rare tumors, in which case the term *carcinosarcoma* is appropriately used. It should be kept in mind that sarcomatoid change may develop in HCC following chemotherapy or transarterial chemoembolization [40].

The characteristic histologic and molecular findings of hepatocellular carcinoma subtypes are summarized in Table 2.

## 6. Precancerous Lesions in Hepatocarcinogenesis

Clonal populations of hepatocytes bearing molecular alterations of the early steps of carcinogenesis may be morphologically recognized in chronically diseased livers as precancerous lesions. These include the following [41]:(i)dysplastic foci (DFs), which are incidentally detected on microscopic examination and measure less than 1 mm in diameter;(ii)dysplastic nodules (DNs), which are larger than dysplastic foci, occasionally measuring over 1 cm in diameter, and may be detected on imaging studies and gross examination

The diagnosis of both DFs and DNs is made by histologic examination. Detection of such lesions is associated with an increased risk of HCC.

DFs are most commonly composed of hepatocytes with small cell change forming a roundish area with increased proliferative activity, as compared to the surrounding parenchyma. Small cell change is characterized by small cell size, increased nuclear–cytoplasmic ratio, mild nuclear pleomorphism and hyperchromasia, and cytoplasmic basophilia [42]. Small cell change of hepatocytes cytologically resembles early HCC. In livers with hereditary hemochromatosis, DFs are characterized by resistance to iron accumulation (“iron-free foci”) [43].

DNs are grossly defined on the basis of comparisons to surrounding liver tissue as “distinctive nodules”. They are most typically distinctive in terms of size, being larger than surrounding cirrhotic nodules [31,44]. However, they may also differ in terms of color (yellow if steatotic, tan-white if fibrotic, dark brown or black if iron-retentive, and green if cholestatic). These lesions are not distinguishable from small HCCs on gross examination. Confirmation that a distinctive nodule is a DN rather than HCC depends on histologic examination. DNs may display cytologic and architectural atypia, but to a degree that is insufficient for a diagnosis of HCC. Most consistently, DNs contain portal tracts, sometimes in a virtually normal distribution, while small, classic HCCs will have destroyed these or pushed them out of the way as they expand. Small classic HCCs will also often display all the histologic features of larger HCCs, such as overt cytologic atypia and thick trabeculae. Distinction between DNs and eHCC is more difficult; this is why eHCC was internationally recognized as an entity only in 2009 [31]. The histologic and immunohistochemical features that are useful for this distinction are discussed below. Sometimes, there are subnodules with features histologically suggestive of HCC within a DN; this is evidence of the DN’s premalignant nature and is also further discussed below.

### 6.1. Low-Grade vs. High-Grade Dysplastic Nodules

DNs are subclassified in two categories, low-grade (LGDNs) and high-grade (HGDNs) [41]. LGDNs are lacking cellular atypia or architectural atypia that would be suspicious for HCC, although they may have large cell change. HGDNs are defined as having cytologic atypia (increased nuclear–cytoplasmic ratio, mild nuclear contour irregularities and hyperchromasia, cytoplasmic basophilia, and small cell change), or architectural atypia (thickened—but less than three cells thick—trabeculae, occasional pseudoglandular structures), which are reminiscent of an emerging HCC but insufficiently extensive to confidently denote a fully progressed HCC. HGDNs may display nodule-in-nodule type of growth, with a distinctive subnodule showing more atypical features. Sometimes the subnodule will merely be more expansile than the surrounding DN parenchyma with increased proliferation producing a “pushing border” at its edges. On occasion, the subnodule will be an overt HCC, displaying stromal invasion into portal tracts or fibrous septa contained within the surrounding DN (Figure 6) [45].

DNs are now understood to represent clonal neoplastic expansions of cells that often develop long before advanced stage liver disease is established [44,46]. They are generally lesions with *low* proliferation compared to surrounding, hyperplastic cirrhotic nodules [47]. (Figure 7). DNs are able to spread, however, because they are also resistant to apoptosis. This resistance gives them a slight survival advantage compared to non-neoplastic hepatocytes in adjacent parenchyma which, in response to the underlying chronic liver disease, have increased turnover [44]. The measure of how slight this advantage must be is that they may take many years to achieve sizes of up to 1.5 cm. DNs’ resistance to the disease affecting the liver as a whole is also evidenced by diminished activation of hepatic stellate cells (HSCs) leading to an absence of scar within the DN or at least diminished scarring compared to the rest of the liver (Figure 7) [48].

### 6.2. Low-Grade Dysplastic Nodules vs. Large Regenerative Nodules

In early studies of DNs in sequential cirrhotic explants, the primary criterion for identifying DNs was a size cutoff (either 0.8 or 1.0 cm, depending on the study). The majority of livers containing DNs have a small number, rarely over 10; however, a subset of liver explants in patients with “macronodular cirrhosis” following either autoimmune hepatitis or hepatitis B had “uncountable” numbers of DNs by this criterion [49]. None of these were HGDN and none of the livers had HCC. Thus, it was clear that sometimes large regenerative nodules (LRNs) can mimic LGDNs.

In resection specimens, histologic distinctions between LGDN and LRN can be counterintuitive. LGDNs are more likely to show relatively preserved, even “normal appearing” parenchymal architecture, while LRNs may show significant disturbances of organization and function, such as variably regenerative or atrophic hepatocytes, large cell change, and hepatocyte injury such as ballooning or cholestasis. Thus, paradoxically, the neoplastic lesions, LGDNs, will appear more like normal liver, while the hyperplastic LRNs will appear reactive and, therefore, abnormal.

If the nodule has some distinctive features that might suggest clonality, this would support a diagnosis of LGDN over LRN. Such changes include diffuse iron or copper accumulation not seen in the surrounding liver or diffuse steatosis, with or without steatohepatitis, in the absence of background fatty liver disease. These findings favor the nodule being a true neoplasm. If one wishes to be more certain, one could do further studies to examine hepatocyte proliferation rates and HSC activation (both low in LGDN and high in LRN) (Figure 7) [47,48]. Moreover, LRNs lack unpaired arteries indicating neoplasia-associated angiogenesis, while LGDNs often have many such vessels (Figure 7) [50,51]. However, in many instances, the distinction between LGDN and LRN may be impossible, particularly in biopsy samples, but even when the whole nodule is present in a resection or autopsy specimen [31].

### 6.3. High-Grade Dysplastic Nodules vs. Hepatocellular Carcinoma

Distinguishing HGDN from well-differentiated HCC can be challenging, especially on needle biopsy material. Recognition of invasive properties, in the stroma or vessels, a hallmark of malignancy (Figure 8), is obviously of paramount importance, but is often difficult to detect. Stromal invasion is the feature distinguishing eHCC from HGDN, and it is suspected when hepatocytes, even some without significant atypia, are present within the stroma of a portal tract or a septum in a large nodule. In such cases, absence of a ductular reaction, confirmed by immunohistochemical stains for cytokeratins 7 or 19, will support the presence of stromal invasion and, therefore, the diagnosis of HCC [45]. Immunohistochemistry can also be useful in biopsy material from nodules where HCC is suspected despite the lack of any evidence of invasion. Immunopositivity of lesional cells for two out of three markers, including glypican-3, glutamine synthetase, and HSP70, is considered diagnostic for HCC (either early or classic), whereas positivity for one or no marker does not resolve the issue of differential diagnosis between HGDN and HCC [52,53].

## 7. Frequently Asked Questions


**
*FAQ 1—Can all hepatocellular neoplasms be definitely classified as either benign or malignant?*
**


Recognizing a hepatocellular proliferation as benign is usually relatively easy, but can be difficult or even impossible in some cases. In livers with advanced chronic disease, the differential diagnosis is basically between high-grade dysplastic nodule and well-differentiated HCC (early or classic). An algorithmic approach to this differential diagnosis has recently been proposed [54]. In livers without chronic disease the difficulties in distinguishing HCA from well-differentiated HCC have long been recognized by experienced liver pathologists and are variably termed in the literature as “atypical hepatocellular adenoma/neoplasm”, “HCA with borderline features”, and “hepatocellular neoplasm with uncertain malignant potential” [1,55]. The worrisome features for the pathologist include architectural abnormalities, such as thickening of liver cell plates, presence of more than occasional pseudoglandular structures, and reticulin disorganization or disappearance, as well as cytological atypia, including presence of small cells, nuclear hyperchromasia, nuclear contour irregularities, nuclear pleomorphism, increased nuclear–cytoplasmic ratio, cytoplasmic basophilia, and presence of more than rare mitotic figures (see Table 1). In such cases, a careful search for features that allow a definite diagnosis of HCC (such as stromal or vascular invasion, trabeculae thicker than three cells, or immunopositivity for the oncofetal proteins α-fetoprotein and glypican-3) is warranted. However, despite careful histopathologic assessment, this differential diagnosis may occasionally remain unresolved. Detection of *TERT* promoter mutation, a marker of approximately 60% of HCCs [17], would be an argument for malignancy in such borderline lesions and holds promise as a diagnostic tool for the future. From a practical point of view, it is currently recommended to indicate this diagnostic difficulty in the report, especially when dealing with a biopsy specimen, in order to trigger appropriate clinical management and/or surveillance.


**
*FAQ 2—Some HCCs arise in completely normal liver. Do these HCCs arise from HCAs?*
**


HCAs are monoclonal neoplasms carrying a risk of malignant transformation reported to be in the range of 4–10%, depending on the series [1]. This percentage is obviously biased because (a) some lesions do not get a biopsy, and (b) many HCAs measuring more than 5 cm are surgically resected or ablated before expressing any potential to evolve to HCC. Since the majority of HCAs arise in normal livers, HCCs arising from and replacing HCAs will also be surrounded by normal hepatic parenchyma, except when an adenomatous rim will still be present at the periphery of the HCC. On the other hand, a minority of HCCs are discovered in normal livers, raising the possibility of a preexisting HCA that cannot be morphologically recognized. None of the immunohistochemical or molecular tools used to diagnose the different subtypes of HCA are useful at this point, because their expression can be modified in malignant lesions; LFABP can be decreased in HCC [56], CRP can be expressed by some HCCs [57], and *CTNNB1* mutations are commonly found in HCC. Therefore, none of these markers can be used for an argument to prove that an HCC arose from an HCA [1]. It is important for the pathologist to check the past medical and imaging history in order to identify clues of a preexisting HCA.


**
*FAQ 3—Do HCAs arise in cirrhotic livers?*
**


Theoretically, the definition of cirrhosis (i.e., a stage in the evolution of chronic liver diseases characterized by scarring and diffuse development of nodules) should not exclude the possibility of HCA of any subtype occurring in cirrhotic livers. However, the clinical context of HCA development is different from chronic liver disease, and pathologists are hesitant to make a diagnosis of HCA in cirrhotic livers. To date, the only HCA subtype that has been reported in livers with cirrhosis is IHCA. Rare IHCAs have been well documented in advanced-stage fatty liver disease, associated with alcohol or metabolic syndrome, with characteristic pathologic, immunohistochemical (overexpression of SAA/CRP), and molecular (different somatic mutations leading to IL6/JAK/STAT pathway activation) features [58,59]. In this context, one must be very cautious and not assert the diagnosis of IHCA only on the basis of immunohistochemical features, since cirrhotic nodules, large regenerative nodules, and dysplastic nodules can overexpress SAA or CRP [58]. Therefore, it is necessary to confirm the presence of a specific IHCA mutation by molecular analysis before reaching a diagnosis of IHCA developing in cirrhotic liver. A fortiori, it is not advisable to affirm this diagnosis on a needle biopsy. As mentioned above, HCC can express CRP, independently from the development in a preexisting IHCA [29].


**
*FAQ 4—Are there any minimum requirements for the use of immunohistochemistry in the diagnosis of HCA?*
**


After confirming that a tumor is an HCA on the basis of H&E-stained sections, it is important to define the subtype, which will determine further patient management. The choice of immunohistochemical stains depends on the pathological features, as demonstrated in Figure 2. If the tumor is highly steatotic, LFABP is mandatory to assert the diagnosis of H-HCA, provided nontumoral liver with normal expression of LFABP is available for comparison. If the tumor exhibits inflammatory features, sinusoidal dilatation, thick arteries, and pseudoportal tracts, CRP and/or SAA is first requested and will lead to the diagnosis of IHCA, if overexpressed. Of note, some H-HCA can be devoid of steatosis and some IHCA can show very little inflammation or show steatosis, which makes both immunostains (LFABP and CRP) useful for the right diagnosis in such cases. On the other hand, in case of a completely characteristic H-HCA, with steatosis and loss of LFABP expression, one can easily conclude that this is the diagnosis. However, in routine practice, even if a step-by-step approach seems to be logical, most of the time, LFABP, CRP, and glutamine synthetase (GS) are determined from the beginning in order to save time and materials. GS is mandatory for three reasons: (1) this marker is very useful to recognize and differentiate the tumoral area from the non-tumoral liver, something not always easy, particularly on biopsy specimens (GS in nontumoral liver is expressed only in a few rows of hepatocytes around the central veins); (2) GS helps to rule out focal nodular hyperplasia (FNH) in case of doubt (absence of classical map-like staining pattern in HCA); (3) GS is the major tool to diagnose *CTNNB1*-mutated HCA with or without associated inflammation allowing the diagnosis of b-HCA and b-IHCA. If GS is strong and diffuse, it means that there is a high level of activation of the β-catenin pathway (most likely due to exon 3 non-S45 mutation). Lower levels of activation of this pathway exist [60], and the pattern of GS expression is a good reflection of this phenomenon, with different immunohistochemical features suggesting different underlying molecular abnormalities, such as at the hotspot S45 of exon 3 or in exon 7/8, resulting in a moderate or low level of β-catenin pathway activation, respectively; in these latter cases, the diffuse CD34 staining in the tumor endothelial cells, except at the peripheral rim, is a good additional argument for the diagnosis [1,2].

It is emphasized that GS is mandatory in all IHCAs in order to reach a diagnosis of b-IHCA, which has the same risk of developing malignant transformation as b-HCA in the case of high-level β-catenin pathway activation. GS immunohistochemistry is much more reliable than β-catenin immunohistochemistry, which is not sensitive enough to identify *CTNNB1*-mutated HCAs. Indeed, this is positive only when GS is strongly expressed and, most of the times, positivity is focal, in a few nuclei. Therefore, there is no need to perform β-catenin immunostaining in HCA subtypes other than b-HCA or b-IHCA.

If LFABP is normally expressed, and stains for CRP and GS are negative, ASS1 is a useful new marker allowing to diagnose shHCA [3,4]. While ASS1 is normally expressed in nontumor liver with a periportal/periseptal pattern (“honeycomb pattern”), overexpression in tumor cells, as compared to nontumor is a requirement in order to make the diagnosis of shHCA. It is important to recognize shHCAs because of their high risk of bleeding. The algorithm (Figure 2) summarizes how to proceed in daily practice.


**
*FAQ 5—Do molecular studies provide any benefit in terms of diagnosis or prognosis of HCA, as compared to standard immunohistochemical stains?*
**


In routine diagnosis, standard immunohistochemical stains (i.e., LFABP, CRP, and GS) are sufficient, most of the time, for the diagnosis of H-HCA, IHCA, b-HCA, and b-IHCA, together representing more than 90% of HCA cases. There is no further benefit to identify inactivation of the *HNF1A* gene by molecular analysis or to search which mutation leads to IL6/JAK/STAT pathway activation, in order to reach a diagnosis of H-HCA or IHCA, respectively.

Concerning the β-catenin pathway, if GS immunostaining is strong and diffuse, it represents evidence that the activation level is high, which means a probable mutation in exon 3, not at the S45 hotspot. In this situation, there is no added value to search which hotspot of exon 3 is mutated for patient management decisions. Indeed, it is well known that these b-HCA/b-IHCA cases have to be resected since they have a high risk of malignant transformation. When the GS immunostaining is heterogeneous or very faint, when the GS-positive peripheral rim is not obvious, particularly in biopsy specimens, or when there are technical problems with immunohistochemistry, molecular methods are useful to search for mutations in exon 3 S45 or exon 7/8, the latter having a very low potential of malignant transformation but a high risk of bleeding, which makes recognition on biopsy material important for further patient management.

Many molecular analyses, such as those concerning *CTNNB1* mutations, can be performed today on formalin-fixed, paraffin-embedded tissue (FFPET), which is easier to obtain than frozen tissue. However, DNA of FFPET may be degraded and, therefore, without value for molecular analysis.

Regarding prognosis of HCA, it has been proposed to search for *TERT* promoter mutations (this is feasible on FFPET) as evidence of malignancy. This would be particularly useful in cases of b-HCA and b-IHCA, when atypical features are present.

In summary, apart from research protocols in referral centers, molecular studies in daily practice add value in terms of subtype diagnosis in b-HCA and b-IHCA, but are not necessary to determine prognosis when resection is mandatory (i.e., men and malignant transformation).


**
*FAQ 6—Should there be different guidelines for the treatment of different types of HCA?*
**


So far, the literature and the existing guidelines [61] indicate that (1) *CTNNB1*-mutated HCAs must be surgically resected or ablated, even if they measure less than 5 cm, (2) HCAs occurring in men also have to be resected or ablated, (3) any HCA measuring more than 5 cm should be resected or ablated, and (4) any HCA that is causing symptoms should be resected or ablated. Emerging evidence from the recent literature suggests that management should be adapted to the subtype more than to the size of the tumors [62]. In cases of adenomatosis, most residual HCAs after resection stabilize or regress, if steatohepatitis and obesity are corrected and/or the OC is discontinued; however, this evolution can take some time [63].

H-HCAs are usually indolent, even if they are large, and they can remain for years without regression and without giving rise to complications, except if they occur in specific clinical contexts, such as vascular liver diseases [64]. Not all b-HCAs and b-IHCAs are at risk of malignant transformation; the risk depends on the type of mutation, with those of exon 3 having the highest risk. On the other hand, shHCAs have a high risk of bleeding, which is clinically significant, even if they are smaller than 5 cm. It is probable that these specificities will guide the establishment of the future guidelines for the management for HCAs. With the aim of building guidelines in mind, it is important to collect standardized clinical and imaging data that led to the clinical management decision in each case [65].


**
*FAQ 7—When should we conclude that an HCA is “unclassified”?*
**


An HCA should be considered unclassified (UHCA) when all other HCA subtypes have been ruled out by currently recommended immunomarkers. Therefore, UHCA should be LFABP-positive, CRP-negative, and SAA-negative, with no abnormal staining of GS and with no abnormal expression of ASS1 (in comparison with the nontumoral liver; see above).

It is recommended, particularly for biopsy specimens, to be cautious with the interpretation because (1) some cases with very light GS staining could be a b-HCA with exon 7/8 mutation and not UHCA, and (2) ASS1 overexpression may be difficult to appreciate in comparison with nontumor liver. In both such situations, it might be advisable to repeat and interpret immunohistochemical stains at referral centers.


**
*FAQ 8—Can we recognize an IHCA when the nontumorous liver is positive for CRP on immunohistochemistry?*
**


It is not rare that nontumorous liver surrounding an IHCA or b-IHCA is CRP-positive, for instance, after portal or arterial embolization, or when there is a severe general inflammatory syndrome with a high level of blood CRP. In such cases, before concluding that a tumor is an IHCA, it is important to be sure that CRP immunopositivity is stronger in the tumor than in nontumorous liver, and to also perform SAA staining for comparison; otherwise, the staining may not be interpretable.


**
*FAQ 9—Is a specialized liver center needed for the management of HCAs?*
**


The clinical management of HCA relies on hepatologists, surgeons, radiologists, and pathologists, sharing their expertise in tumor board meetings. Imaging techniques are reliable to identify most cases of the H-HCA and IHCA subtypes, provided the radiologist has some experience and uses specific techniques. By contrast, specific recognition of b-HCA, b-IHCA, and shHCA by imaging is still under investigation. In their routine practice, with the help of immunohistochemistry, pathologists can also provide a diagnosis of H-HCA, IHCA, and b-(I)HCA (in case of a strong diffuse GS staining). b-(I)HCA with other patterns of GS staining and shHCA are less well known, and interpretation of immunohistochemistry can be difficult requiring confirmation by molecular methods. In clinical centers where HCAs are rare, referring patients to specialized centers will improve diagnosis and decision making and, for the rarer subtypes, will also help to foster cutting-edge guidelines for patient management.


**
*FAQ 10—Should the HCC grade (i.e., degree of differentiation) be reported in biopsy and in surgical specimens? What is the best grading system?*
**


Similar to other carcinomas, HCCs are graded as well, moderately, or poorly differentiated. The grade is a marker of prognosis, and has been found to predict patient survival and disease-free survival after both surgical resection and liver transplantation [66,67,68]. Therefore, HCC grade represents useful information that should be included in pathology reports. However, HCC often displays variable differentiation in different parts of the tumor. While prognosis would be expected to be primarily related to the least differentiated component of the neoplasm, knowledge of the existence of other components may be useful for the assessment of additional specimens, such as those obtained at later dates from metastatic sites. On the other hand, biopsy specimens may not be representative of the entire range of differentiation present in any given HCC, due to sampling error. Nevertheless, a study has found significant correlation between grade assessment of the biopsy and subsequent surgical specimen of HCC arising in cirrhotic patients [69]. Therefore, HCC grade should be reported in biopsy specimens, with the understanding that it may not always be entirely representative.

Various grading systems for HCC have been devised over the years. What is important for practical purposes is reproducibility of grading and clinical usefulness of the system. While the four-tiered Edmondson–Steiner system has been widely used in clinical studies, a three-tiered system is currently favored for daily practice [30]. It is hoped that adequate description of the characteristic features of each grade will result in high reproducibility among pathologists.


**
*FAQ 11—Which is the best immunohistochemical panel to assure hepatocellular origin of a malignancy?*
**


The most commonly used immunohistochemical markers of HCC are those mentioned in Section 5, i.e., arginase-1, HepPar1, glypican-3, and α-fetoprotein. Arginase-1 is the most sensitive and specific marker for HCC, staining over 90% of cases, while nonhepatocellular tumors are rarely positive for this marker [70]. HepPar1 stains most well differentiated HCCs, but less than 50% of poorly differentiated ones. Furthermore, various adenocarcinomas may occasionally be positive for HepPar1, particularly those from the small intestine, the normal enterocytes of which also uniformly express the antigen [71]. Glypican-3 is positive in 65–80% of HCCs, more often in poorly differentiated than well differentiated tumors; however, glypican-3 may also be positive in a variety of other malignancies, such as carcinomas from other sites, melanoma, and germ cell tumors [70]. α-Fetoprotein has low sensitivity for HCC (<50%) and may be expressed in germ cell tumors and rare other malignant neoplasms. In addition to these markers, HCCs arising in patients with chronic hepatitis B may occasionally be positive for HBsAg, an uncommon but most specific finding. Lastly, in situ hybridization for albumin mRNA can be very useful in distinguishing HCC from other malignancies but is available in a limited number of institutions.

Immunohistochemical stains for carcinoembryonic antigen utilizing polyclonal antibodies (pCEA) often provide a canalicular pattern of staining that is useful for HCC diagnosis. However, poorly differentiated HCCs often lack this pattern and may, instead, display membranous or even cytoplasmic staining, similar to that of adenocarcinomas. Immunohistochemical stains for CD10 often demonstrate in HCC a similar canalicular pattern of staining as that seen with pCEA. Again, this is usually absent in poorly differentiated HCC. An example of establishing the diagnosis of HCC with the aid of immunohistochemical stains is shown in Figure 9.

However, demonstrating the hepatocellular nature of a poorly differentiated carcinoma may be difficult in some cases. This is especially true in biopsy specimens with limited material, taking into account that immunopositivity of tumor cells for the markers mentioned above may be focal, resulting in false-negative findings. Clinicopathologic correlation taking into account all the clinical, imaging, and pathologic findings will be essential in such cases. Appropriate additional markers for other tumors should also be included, as per the differential diagnosis in each particular case.

On the other hand, if a well-differentiated carcinoma consists of what appear to be hepatocytes, then an appropriate panel would include arginase-1, HepPar1, pCEA (or CD10), and glypican-3, as α-fetoprotein has very low yield in well-differentiated HCC. If the lesion is truly HCC, any one marker may be positive, or a pair or more may show staining. However, stains for mimics of well differentiated HCC—particularly renal cell carcinoma, adrenal cortical carcinoma, neuroendocrine tumors, and follicular thyroid carcinomas—should also be considered [72]. Again, clinicopathologic correlation is essential.


**
*FAQ 12—Are there any HCC subtypes that need to be specified on histologic diagnosis?*
**


The importance of HCC subtyping is related to differences in clinical correlations, prognosis and treatment among subtypes. A subtype that is important to identify is the fibrolamellar HCC, which is characterized by young patient age, lack of underlying liver disease and tendency to metastasize to hilar lymph nodes. Therefore, the mainstay of therapy is surgery, including regional lymphadenectomy. The recent discovery of a characteristic somatic gene fusion in fibrolamellar HCC (see Section 4) allows optimism for future development of targeted therapies for nonresectable tumors. Other subtypes are worth identifying because of prognostic differences, as compared to average; for example, the macrotrabecular massive subtype and the neutrophil-rich subtype are associated with worse prognosis, while the clear cell subtype with better prognosis.

It is emphasized that HCC subtyping is a work in progress. With the exception of the fibrolamellar subtype, HCC had been regarded until recently as a tumor of bleak prognosis; therefore, there was little interest in subtyping. This view is now changing because of the availability of surveillance programs and the hopes for new therapies based on molecular profiling. Therefore, correlation of the clinical, pathologic, and molecular features in large series of patients with the various subtypes of HCC may allow a more individualized approach for treatment. As more data become available, the importance of subtyping will likely increase. Histologic examination is the basis for HCC subtyping; therefore, criteria of each subtype should be clear and reproducible.


**
*FAQ 13—Do immunohistochemical stains or molecular studies provide any actionable items for HCC? Do molecular studies provide any added value in terms of diagnosis or prognosis of HCC, as compared to standard immunohistochemical stains?*
**


The main use of immunohistochemical stains in cases of suspected HCC is to confirm the diagnosis and rule out other neoplasms. However, there is also a stain that has been found to be of prognostic significance in HCC; cytokeratin 19-positive HCCs have higher recurrence rates than usual, as well as higher resistance to locoregional therapies [73,74,75]. It should be kept in mind that cytokeratin 19 is positive in a variety of adenocarcinomas, including cholangiocarcinoma; therefore, it is used as a prognostic, but not as a diagnostic marker for HCC.

Molecular methods have been used extensively in recent years to identify the molecular changes occurring in hepatocarcinogenesis, including potential targets for treatment. Substantial molecular data have been accumulated, and several molecular classifications have been proposed on the basis of a correlation of clinical, pathologic, and molecular data [22,26,27,28,29]. However, these classifications have not yet found their way to clinical practice. As a result of these studies, most HCCs can now be grouped into two classes [25,76]: (i) the proliferation class, which is etiologically related to HBV infection, and displays molecular and histologic features associated with aggressive clinical behavior; (ii) the nonproliferation class, which is etiologically related to HCV infection or alcohol, and displays features associated with better clinical outcome. HCCs of the proliferation class are characterized by *TP53* mutations, chromosomal instability, and activation of various oncogenic pathways, tend to be poorly differentiated, and are associated with high serum α-fetoprotein. On the other hand, HCCs of the nonproliferation class often have *CTNNB1* mutations and a gene expression profile resembling that of normal hepatocytes. These tumors tend to be better differentiated and with lower incidence of vascular invasion than those of the proliferation class.

The characteristic molecular changes of the various HCC subtypes are summarized in the section on diagnosis of hepatocellular carcinoma (see Table 2). Although not routinely used in daily diagnosis, detection of these changes can be used in support of the diagnosis. For instance, detection of the gene fusion *DNAJB1–PRKACA* can confirm the diagnosis of fibrolamellar HCC.


**
*FAQ 14—What should pathologists know and do about combined hepatocellular-cholangiocarcinomas (cHCC–CCA)?*
**



*FAQ14.1. Tissue Diagnosis of cHCC–CCA*


A tissue diagnosis of a primary cHCC–CCA is straightforwardly made by routine hematoxylin–eosin stains; immunostains for markers of hepatocyte or cholangiocyte differentiation are merely confirmatory [30,77]. The presence of stainable hepatocyte markers in glandular epithelium (e.g., arginase-1, HepPar1, α-fetoprotein, glypican-3, and albumin mRNA) or, conversely, of cholangiocyte markers in HCC (e.g., keratins 7 and 19, and EpCAM) are not proof of cHCC–CCA given the possibilities of aberrant gene expression in any malignancy [77]. Differentiated components of cHCC–CCA may be located in distinct areas of a tumor, or they may be intimately intermingled throughout the lesion. Boundaries between the components may be sharply defined or indistinct. There are, as yet, no definitive cutoffs for a percentage requirement for the presence of the component. A minute component of intrahepatic CCA (iCCA) within an otherwise clear HCC is sufficient to call it cHCC–CCA and vice versa.

A common pitfall of diagnosis is when cHCC–CCA is suspected on radiographic grounds, but only one element is present in the biopsy specimen. In this case, the pathologist must be careful to comment on the limitations of biopsy. Small biopsy specimens may sample only one component of such a heterogeneous tumor; the absence of the other component does not exclude cHCC–CCA and a formal statement to that effect in the pathology report is important. On the other hand, metastatic lesions associated with a primary cHCC–CCA may comprise either component alone or mimic the cHCC–CCA appearance of the primary tumor. Biopsy specimens from metastatic lesions must be cautiously interpreted in this light [78].

Another pitfall for diagnosis can occur when there appear to be two separate mass lesions in the liver that are merging together. An HCC and a separate, but simultaneous CCA may grow into each other forming a “collision tumor”, particularly in chronic liver diseases that predispose to both malignancies. Each of these tumors should be assessed pathologically as independent entities.

Lastly, a very rare variant of “intermediate cell type” of cHCC–CCA notably breaks all the rules; its tumor cells appear morphologically intermediate between hepatocytes and cholangiocytes and do not show typical growth patterns of either HCC (e.g., trabeculae and pseudoglandular structures) or iCCA (e.g., mucin-producing glands, tubules, and signet ring cells), often appearing homogeneous throughout. Dual differentiation in these tumors is, indeed, at the cellular level, with each cell showing combined hepatocyte and cholangiocyte marker expression [30].

It should also be noted that, while cHCC–CCA may present, de novo, as a primary hepatic malignancy, it has also been seen to emerge from HCCs that have undergone loco-regional treatments [79]. It has been suggested that hypoxia of surviving tumor cells after transarterial chemoembolization leads to expression of proteins, such as EpCAM and cytokeratin 19 [80]. Such adaptive changes might explain emergence of cHCC-CCA from a treated HCC, although the possibility that there was a previously undetected minor component of CCA originally is difficult to exclude. In any case, while this occurrence seems uncommon, it should be considered when tumor recurs post treatment, particularly if imaging features no longer show classic features of HCC, alone.

Subpopulations of tumor cells in cHCC–CCA may have what has been described as a “stem-cell appearance”, i.e., small cells with high nuclear–cytoplasmic ratio, sometimes arrayed with larger hepatobiliary cells in what appear to be lineage relationships like those seen in ductular reactions in diseased or injured liver. While subclasses of “stem-cell tumors” were characterized in the 2010 edition of the WHO “Blue Book” [81], the more recent edition [82] has eliminated the term as a diagnostic category in all primary liver cancers. Nonetheless, the question of its importance remains uncertain, and it has been recommended that the presence of “stem-cell features” be noted in the pathology report of tumors that contain them [83].


*FAQ14.2. Pathology–Radiology Collaboration for cHCC–CCA*


A biphasic radiographic appearance of a lesion may help a pathologist avoid missing the opportunity for including cHCC–CCA in the differential diagnosis when only one component is sampled in a biopsy specimen [84]. However, in multidisciplinary conferences for liver malignancies, clinicians and radiologists may miss clues to cHCC–CCA given their rarity, but the attentive pathologist can help guide radiologists toward how to best sample a lesion for successful complete diagnosis.

In early-stage liver disease or in sporadic tumors in which there is no predisposing hepatic disease, cHCC–CCA may appear biphasic, with separate areas showing typical features of HCC or iCCA, or they may merely be atypical, without imaging characteristics specific for either, thus being more suggestive of metastasis [84]. In advanced-stage liver disease, in which the Liver Imaging Reporting and Data System (LI-RADS) classification is applicable, radiologists may label a lesion LIRADS-M because unusual features suggest a metastasis, but the lesion may actually be a cHCC–CCA [85,86]. Alternatively, they may recognize that one or more parts of the lesion have an LIRADS-5 (diagnostic for HCC) appearance and, thus, label the whole lesion with that designation, even though some areas appear distinct [84,85]. In both these settings, a pathologist who is attentive to radiographic descriptions that might hint at cHCC–CCA may save the day.


*FAQ14.3. Molecular Pathology and Treatment Implications of the Diagnosis of cHCC–CCA*


Molecular studies support that these tumors may sometimes derive from a malignantly transformed hepatobiliary stem/progenitor cell or from de-/redifferentiation of malignantly transformed hepatocytes or cholangiocytes, and that they may be more like iCCA, more like HCC, or intermediate between them. All of these data confirm that they are certainly, at least to some degree, heterogeneous in origin and in behavior [87]. On the other hand, comparison of clinicopathological characteristics of cHCC–CCA with regard to the newest WHO classification [82] supports its relevance and that cHCC–CCA has intermediate survival between HCC and iCCA, if not actually tilting toward the dire outcomes for iCCA [87,88]. Given the propensity for early and distant spread of CCA components along lymphatic and perineural pathways that are typical of iCCA itself, it is no surprise that clinical outcomes after resection are, overall, worse than for HCC. On the other hand, if transplanted within the Milan criteria, cHCC–CCA showed similar overall survival to HCC after transplantation [89].

Unfortunately, the rarity of these tumors has interfered with the performance of randomized clinical trials. There is a paucity of data regarding immunotherapies [90], although studies suggest that at least some cHCC–CCA should be responsive to these types of treatment [91,92]. Broad genomic profiling of malignancies for actionable mutations specific to each case is currently the most likely path to any possible clinical benefit [93].


**
*FAQ 15—I do not work in a transplant center and am so unlikely to ever see a dysplastic nodule specimen. Do I need to know about them? If so, why?*
**


Yes! One needs to know about them! Currently, screening for emergence of malignancy in chronic liver disease depends largely on radiographic criteria defined by the Liver Imaging Reporting and Data System (LI-RADS) classification [94]. Pathologists may find themselves involved in liver tumor multidisciplinary conferences in which radiologists will report distinctive nodules that are subclassified into LIRADS-1 through 5, with a higher score indicating the higher confidence that a lesion is an actual HCC (Figure 7) [95]. The repetition of the phrase “distinctive nodule” is not a coincidence; LI-RADS was formulated to reflect the pathologic understanding of DNs as neoplastic and often premalignant lesions.

A classification of LIRADS-5 is so specific for HCC that, in most medical centers, it is sufficient for diagnosis without confirmatory biopsy. LIRADS-1 lesions are considered likely to be benign, probably merely large regenerative nodules. LIRADS-2 through 4 probably reflect LGDN through HGDN (although direct pathology–radiology correlations for these have not been reported) [94,95].

It is not necessary for the pathologist to know the full and subtle criteria for the LIRADS classifications, but it is vital to know what lesions each designation may reflect, thus enabling the pathologist to carefully guide the clinicians and radiologists in terms of follow-up screening or treatment of the patient. It is worth knowing, however, that the increasing stages of the LI-RADS classification probably reflect the changes in vascular supply. Regenerative nodules and LGDNs have mostly intact portal vein blood flow with little increase in arterialization. However, the increasing ratio between enlarging arterial blood flow (angiogenesis inside truly neoplastic LGDN and HGDN) and the diminishing blood supply (as portal tracts are degraded or pushed to the sides) result in the characteristic LIRADS features (Figure 7).

Examples are provided below.

**LIRADS-1**: The pathologist may advise the clinical team that, while this lesion is probably just a regenerative nodule, the possibility that it is a DN, possibly even an HGDN, is not excluded. Repeat screening at a shorter time interval may be warranted. If the lesion disappears, it was probably large regenerative nodule that underwent involution or further scarring that eliminated its distinctive appearance on imaging. If it does indeed disappear, return to normal surveillance screening is reasonable.**LIRADS-2 or -3**: These lesions are more likely to be DNs, either LGDN or HGDN. Repeat imaging should be performed more frequently. If the nodule disappears, it was probably regenerative. If it persists, then it may be LGDN or HGDN and the patient is considered at higher risk for HCC and should return more frequently for imaging. If the lesion progresses upward in LIRADS score, it is probably a DN giving rise to an HCC. Continued imaging or ablation may be considered depending on the clinical circumstances.**LIRADS-4**: These lesions are probably an HGDN, possibly with an emerging focus of HCC, or possibly a small HCC. Continued imaging or ablation may be considered depending on clinical circumstances.**LIRADS-5:** This feature is diagnostic for HCC. While a biopsy is not necessary for diagnosis, oncologists are increasingly requesting a pre-ablation biopsy for molecular studies to inform future treatments if the lesion is resistant to ablation or if there is post-treatment (i.e., ablation, resection, or transplant) recurrence.**LIRADS-M:** These are lesions without typical imaging features of HCC, but highly suspicious for malignancy. The pathologists may aver that the lesion could be metastatic (M), but they could also be iCCA or cHCC–CCA. Such lesions probably require biopsy for diagnosis.

With regard to cHCC–CCA, it behooves the pathologist to watch for lesions described by the radiologist as “complex” or as having an isolated part with typical LIRADS-5 features while the other parts of the lesion are atypical for HCC [84,85,86]. Because cHCC–CCA may have distinct regions of the tumor that are either HCC or CCA, in such tumors, some regions will show LIRADS-5 changes, while others will not. This situation is one in which the pathologist can make a decisive difference, recommending targeted biopsies of *both* the LIRADS-5 and the atypical areas. iCCA’s worse prognosis and different treatment implications make it important to diagnose as early as possible in the treatment course. The pathologist may be the only person in the room sensitive to this uncommon cancer. Alerting the clinical team to make sure that cHCC–CCA has been completely evaluated may be crucial for saving the life of this patient or preventing an inappropriate transplant for an incurable malignancy.


**
*FAQ 16—When should tumors with imaging features of HCC be biopsied?*
**


It is currently believed that the findings which define LIRADS-5, i.e., those lesions with all imaging features of HCC by imaging, do not require confirmatory biopsy. However, ablative treatment of these lesions is then likely, and tissue for molecular analysis for determining possible targeted therapies will not be available. For this reason, there may be a shift in clinical practice toward biopsy of LIRADS-5 HCCs in the near future, not for diagnosis or prognosis, but for determination of suitable targeted therapies in the event of recurrence. Subtyping of HCC might also be found to be relevant in the near future to guide treatment decisions, also with the support of artificial intelligence [96].

As noted above, if part of a lesion displays typical HCC imaging features, but other parts do not, the possibility of a cHCC–CCA cannot be excluded. In addition, even rarer HCC variants could appear in combination with classic HCC, such as the sarcomatoid one, which might have nontypical imaging features. In such cases, biopsy of both the classic HCC component and of the nontypical component is warranted.

## Figures and Tables

**Figure 1 cancers-14-03670-f001:**
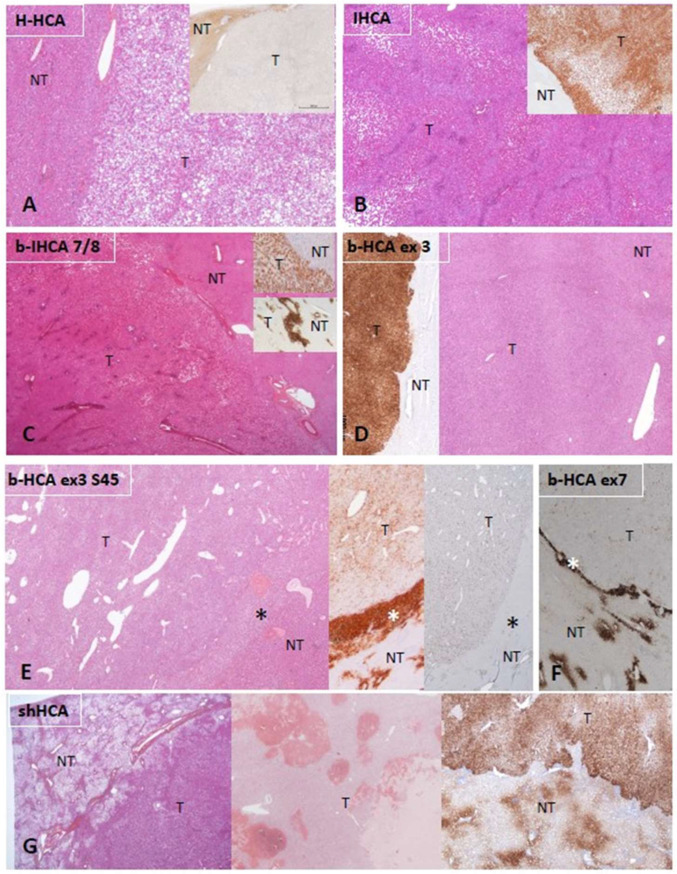
Characteristic histologic and immunohistochemical features of HCAs. (**A**) H-HCA: The tumor (T) appears highly steatotic on H&E with a complete lack of LFABP by immunohistochemistry (insert), contrasting with the normal expression in the nontumorous liver (NT). (**B**) IHCA: The tumor (T) exhibits sinusoidal dilatation on H&E with a strong CRP expression by immunohistochemistry (insert), sharply demarcated from the nontumorous liver (NT). (**C**) *CTNNB1* exon 7/8 mutated b-IHCA: This tumor exhibits a classical appearance of IHCA (sinusoidal dilatation, numerous thick arteries, and strong expression of CRP (top insert); in addition, GS is very faint in the tumor but with a strong GS rim between tumor (T) and nontumorous liver (NT) (bottom insert); molecular analysis identified a mutation on *CTNNB1* exon 7/8 (see [2]). (**D**) *CTNNB1* exon 3 mutated b-HCA: This tumor (T), which is not well delimited from the nontumorous liver (NT) on H&E, exhibits a strong and diffuse GS expression (left insert), identifying a high level of activation of the β-catenin pathway (large deletion on exon 3). (**E**) Exon 3 S45 mutated b-HCA: This tumor (T) exhibits numerous irregular vessels below the rim (asterisk) that separates T from nontumorous liver (NT); heterogeneous expression of GS is seen in T, whereas a strong GS expression characterizes the rim (middle insert); a corresponding diffuse CD34 immunostaining is seen in the endothelial cells of T, with no CD34 expression in the rim (asterisk) (right insert) (see [2]). (**F**) Exon 7 mutated b-HCA: GS is very faint in the tumor (T), and a thin GS rim (asterisk) separates T from the nontumorous liver (NT); molecular methods identified a β-catenin exon 7 mutation. (**G**) shHCA: This tumor developed in a highly steatotic nontumorous liver (NT, left picture) and exhibits focally large hemorrhagic foci (middle picture); ASS1 immunohistochemistry shows an overexpression in the tumor (T), in comparison with the nontumorous liver (NT), in which its expression is restricted to the periportal/septal zones (right picture). *Abbreviations:* H-HCA, *HNF1A*-mutated hepatocellular adenoma; IHCA, inflammatory HCA; b-IHCA, β-catenin-mutated inflammatory HCA; b-HCA, β-catenin-mutated HCA; shHCA, sonic hedgehog-activated HCA; LFABP, liver fatty-acid-binding protein; CRP, C reactive protein; GS, glutamine synthetase.

**Figure 2 cancers-14-03670-f002:**
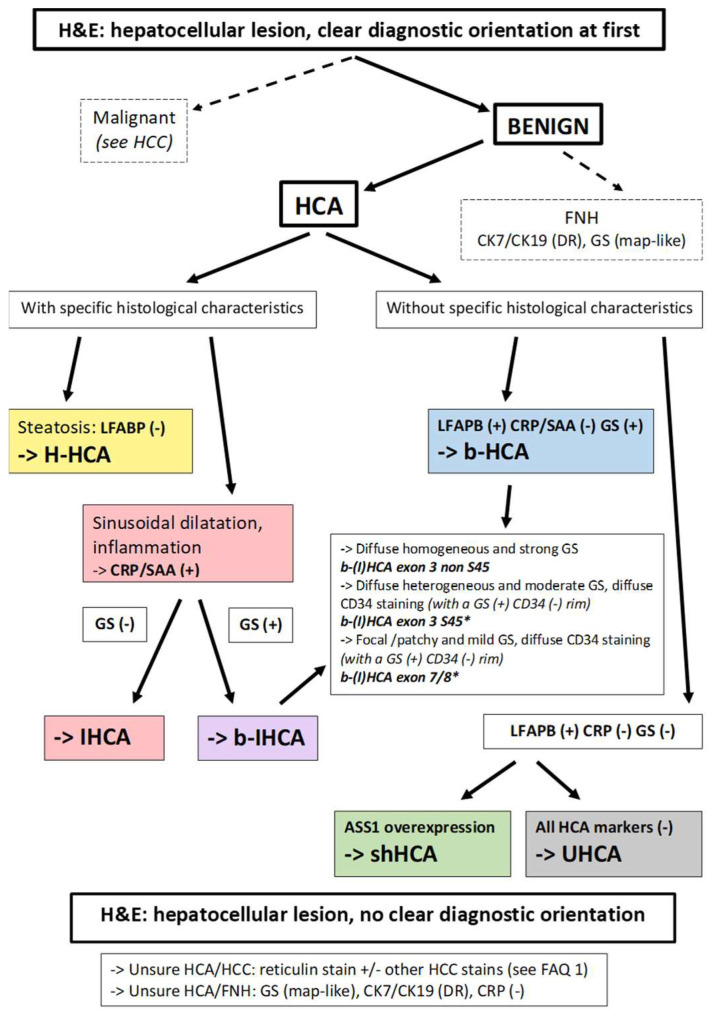
Diagnostic algorithm for HCAs. From a practical point of view, most of the cases are easily recognized as benign or malignant, but some are not. In the situation of an obvious HCA, if there is steatosis, with LFABP (−) and GS (−), there is no need to perform further IHC staining; it can be concluded that the tumor is an H-HCA. If an HCA shows sinusoidal dilatation and inflammation, with LFABP (+) and GS (−), it is mandatory to perform CRP and/or SAA immunostaining in order to diagnose an IHCA. GS immunostaining is mandatory in all IHCAs in order to diagnose a b-IHCA. Different patterns of GS staining exist, linked to the type of underlying mutations (see [2]). If LFABP is positive and all other markers are negative, then an overexpression of ASS1 will lead to the identification of a shHCA, whereas, if it is not overexpressed, it is an UHCA. * Importantly, the GS(+)/CD34(−) rim can be irregular or discontinuous and is usually better represented in b-HCA than in b-IHCA. Its recognition on biopsies can be challenging (see [2]). In case of an uncertain diagnosis, HCA versus HCC or HCA versus FNH, additional histochemical and immunohistochemical stains are needed. The differential diagnosis of HCA versus HCC is discussed in FAQ 1. Reticulin stain might help to recognize alterations of the framework, although it is not a strict feature. Cytokeratin 7 and cytokeratin 19 stains help to recognize ductular reaction, and GS has a specific map-like pattern in FNH. *Abbreviations:* HCA, hepatocellular adenoma; H-HCA, *HNF1A*-mutated HCA; IHCA, inflammatory HCA; b-HCA, β-catenin-activated HCA; b-IHCA, β-catenin-activated and inflammatory HCA; shHCA, sonic hedgehog-activated HCA; UHCA, unclassified HCA; HCC, hepatocellular carcinoma; FNH, focal nodular hyperplasia; LFABP, liver fatty-acid-binding protein; CRP, C reactive protein; SAA, serum amyloid A; GS, glutamine synthetase; ASS1, argininosuccinate synthase; CK7, cytokeratin 7; CK19, cytokeratin 19; DR, ductular reaction.

**Figure 3 cancers-14-03670-f003:**
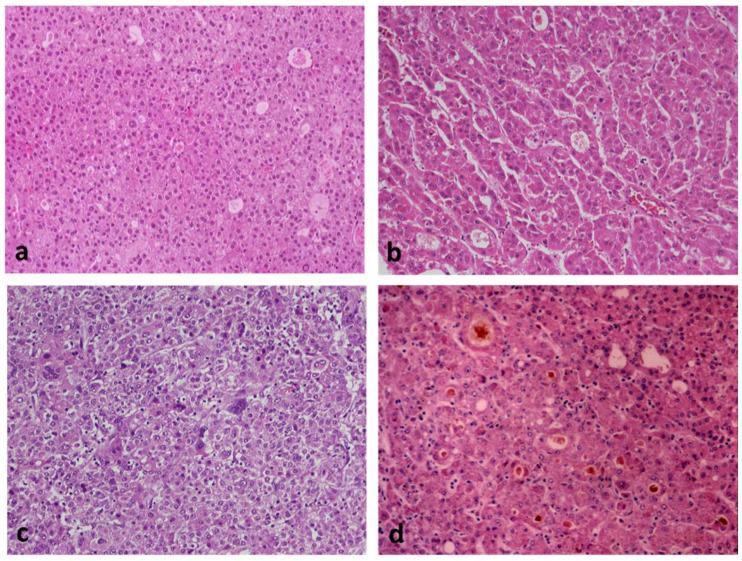
Degrees of differentiation in HCC: (**a**) This well-differentiated HCC consists of neoplastic cells resembling hepatocytes, which are arranged in trabeculae and pseudoglandular structures. (**b**) As compared to (**a**), this moderately differentiated HCC displays an increased nuclear–cytoplasmic ratio, larger nuclei with prominent nucleoli, and increased cytoplasmic basophilia. (**c**) This poorly differentiated HCC is characterized by marked tumor cell pleomorphism, including multinucleated cells; the architecture is trabecular and compact. (**d**) Bile production by neoplastic cells, often in pseudoglandular structures, as illustrated here, is a diagnostic feature of HCC.

**Figure 4 cancers-14-03670-f004:**
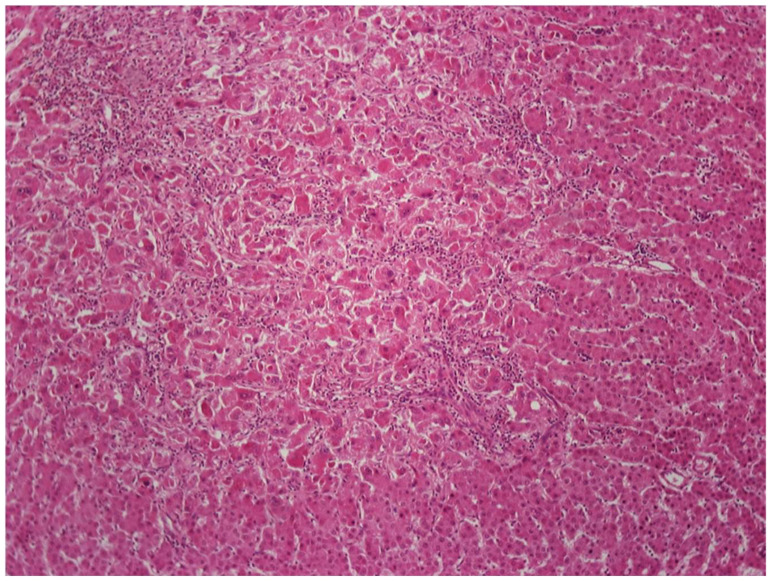
Classic HCC arising within early HCC (right and lower parts of the picture). Note the small unpaired arteries (right middle and lower part of the picture).

**Figure 5 cancers-14-03670-f005:**
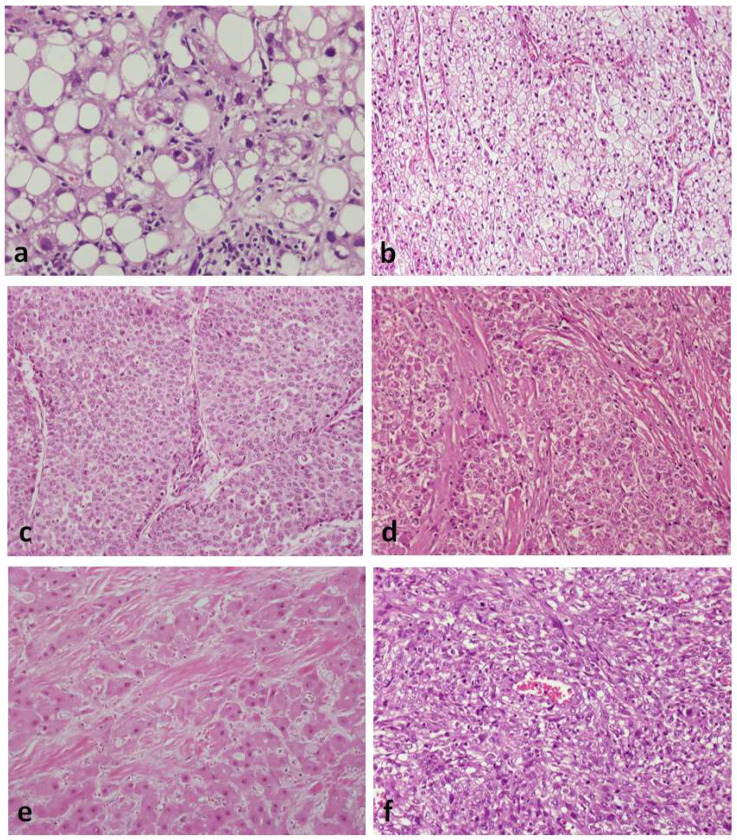
Examples of HCC subtypes: (**a**) steatohepatitic; (**b**) clear cell; (**c**) macrotrabecular; (**d**) scirrhous; (**e**) fibrolamellar; (**f**) sarcomatoid.

**Figure 6 cancers-14-03670-f006:**
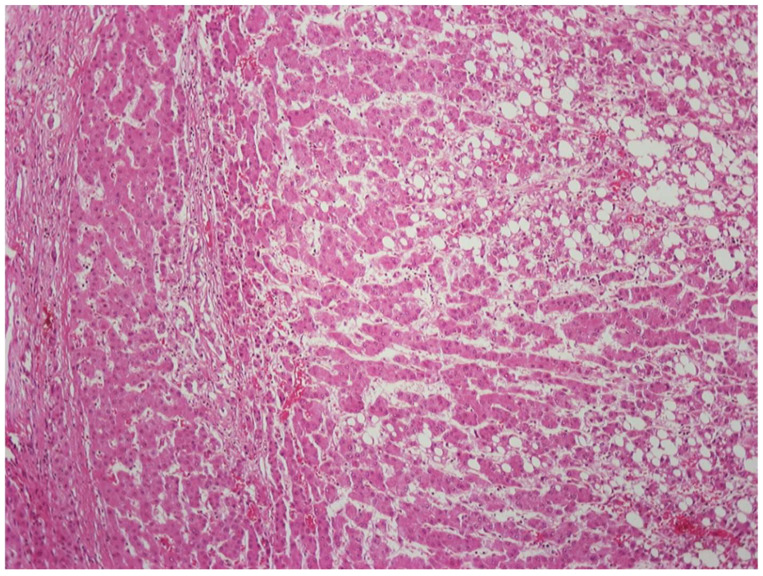
HCC (central and right part of the picture) arising in dysplastic nodule (**left part**). The tumor has features of early HCC (**central part**) and classic HCC with steatosis (**right part**).

**Figure 7 cancers-14-03670-f007:**
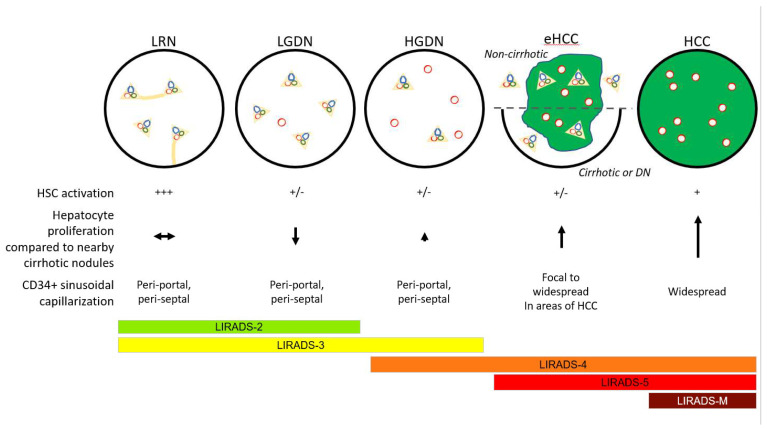
Important histologic features in hepatocellular nodules emerging in chronically diseased livers, and their LIRADS correlation. *Abbreviations:* LRN, large regenerative nodule; LGDN, low-grade dysplastic nodule; HGDN, high-grade dysplastic nodule; eHCC, early hepatocellular carcinoma; HCC, classic (progressed) hepatocellular carcinoma; HSC, hepatic stellate cell.

**Figure 8 cancers-14-03670-f008:**
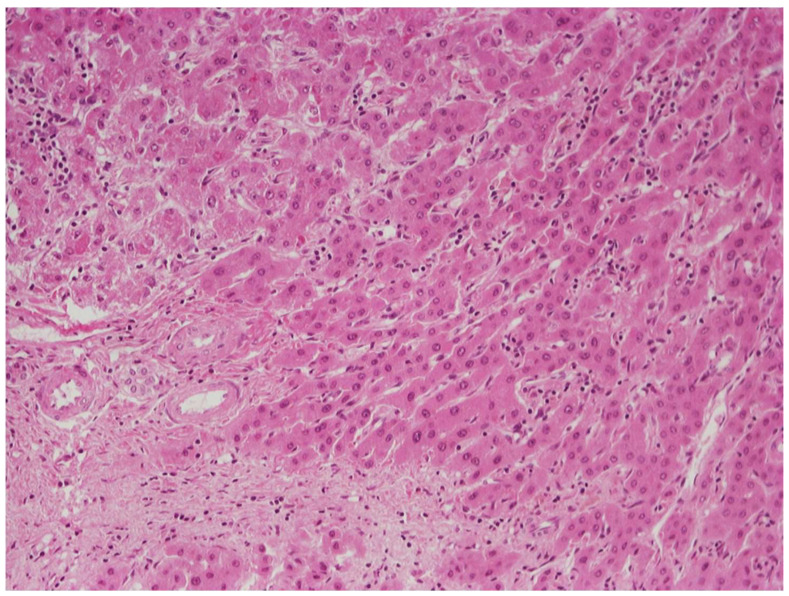
Well-differentiated HCC invading portal tract and fibrous septum in liver with advanced stage chronic hepatitis C. Note the absence of ductular reaction.

**Figure 9 cancers-14-03670-f009:**
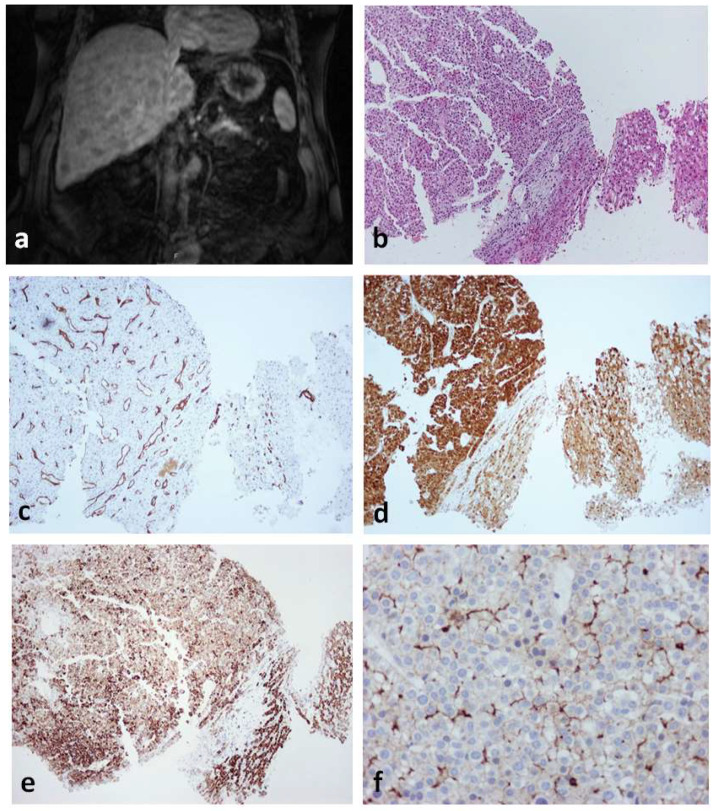
Abdominal MRI (**a**) and guided liver biopsy specimen (**b**–**f**) from a 53 year old man with breathing difficulty and history of metabolic syndrome, including obesity, type 2 diabetes mellitus, and hyperlipidemia. (**a**) There was marked hepatomegaly with innumerable, scattered nodules, measuring up to 3 cm, suggesting metastatic disease. However, needle biopsy revealed HCC. In this limited biopsy material, polygonal tumor cells appeared to be arranged in a compact sheet (**b**, **left side**); nevertheless, immunohistochemical stain for CD34 (**c**), highlighting the endothelial cells, demonstrated trabecular architecture. Further stains showed positivity of tumor cells for arginase-1 (**d**) and HepPar1 (**e**). Adjacent hepatocytes (**d**,**e**, **right side**) are also positive for these markers, serving as “internal controls”. Tumor cells also displayed a canalicular pattern of staining with pCEA (**f**), as well as positivity for glypican-3 (not shown).

**Table 1 cancers-14-03670-t001:** Comparison of etiology, pathogenesis, and diagnostically useful histopathologic features of hepatocellular neoplasms.

	Hepatocellular Adenoma	Hepatocellular Carcinoma
**Etiology and Pathogenesis**		
Chronic liver disease	Usually absent	Usually present
Molecular changes	Four specific morpho-molecular	Large variety of mutations
	subtypes, including the following:	affecting a number of signal
	- H-HCA: *HNF1A*-inactivating mutations	transduction pathways;
	- IHCA: mutations activating IL6/JAK/STAT	most frequent mutations
	- b-HCA, b-IHCA: *CTNNB1*-activating mutations	Involve TERT promoter
	- shHCA: *INHBE–GLI1* gene fusion	
**Tumor architecture**		
Thickness of cell plates	1–2 cells	Variable
Pseudoglandular structures	Absent or few	Absent or present
Reticulin fibers	Preserved or focally disorganized	Decreased, disorganized
Invasive growth in stroma or vessels	Absent	Present
**Cytologic features**		
Small cell size	Uncommon	Sometimes present
Nuclear hyperchromasia	Uncommon	Commonly present
Nuclear contour irregularities	Uncommon	Commonly present
Nuclear pleomorphism	Uncommon	Commonly present
Nuclear–cytoplasmic ratio	Usually normal	Often increased
Cytoplasmic basophilia	Usually absent	Commonly present
Mitotic figures	Absent or rare	Often present
**Nonlesional hepatic parenchyma**		
Evidence of cirrhosis	Absent (rarely present in IHCA)	Present or absent
**Positive immunohistochemical staining**		
Alpha-fetoprotein	Absent	Present or absent
Glypican-3	Absent	Present or absent

**Table 2 cancers-14-03670-t002:** Characteristic histologic and molecular findings of hepatocellular carcinoma subtypes.

Subtype	Characteristic Histologic Findings	Characteristic Molecular Findings
Steatohepatitic	Features simulating steatohepatitis (macrovesicular steatosis, inflammation, ballooned cells, Mallory–Denk bodies, and pericellular fibrosis)	IL6/JAK/STAT pathway activation
Clear cell	Glycogen accumulation in tumor cells	None to date
Macrotrabecular massive	Thick trabeculae (>6 cells thick)	*TP53* mutations, *FGF19* amplifications
Scirrhous	Diffuse fibrosis	*TSC1/TSC2* mutations
Chromophobe	Light staining cytoplasm, mostly bland nuclei, occasional large atypical nuclei; cystic spaces with serum-like material	Alternative lengthening of telomeres
Fibrolamellar	Large polygonal cells with abundant eosinophilic cytoplasm, large nuclei and prominent nucleoli; pale bodies; lamellar fibrosis; immunopositivity for cytokeratin 7 and CD68	*DNAJB1–PRKACA* gene fusion
Neutrophil-rich	Abundant intratumoral neutrophils	G-CSF production by neoplastic cells
Lymphocyte-rich	Abundant intratumoral lymphocytes	None to date
Sarcomatoid	Spindle cell morphology	None to date

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
