# Peer review of "Etiology, Pathogenesis, Diagnosis, and Practical Implications of Hepatocellular Neoplasms"

_cancers, 2022, doi:10.3390/cancers14153670_

Round 1
Reviewer 1 Report
This review summarizes comprehensive aspects of etiology, pathogenesis, and histologic diagnosis of hepatocellular adenoma (HCA) and hepatocellular carcinoma (HCC). In addition, this review recommends practically useful answers and solutions to a range of frequently asked clinical questions. It is believed that this review will be valuable for surgeons and researchers in understanding HCA.
Some aspects of improvement are recommended for this review including:
1. A table to summarize and compare the etiology, pathogenesis, and diagnosis between HCA and HCC is highly recommended.
2. Future developments for diagnosis and treatments for HCA
Author Response
We thank Reviewer 1 for the kind comments. Indeed, our purpose was to write a useful article; therefore, we are very pleased that Reviewer 1 considers our article to be valuable.
- We have provided a table comparing the etiology, pathogenesis, and diagnosis of HCA and HCC, as recommended by Reviewer 1 (Table 1).
- Future developments for diagnosis and treatment for HCA are covered in FAQs 5 and 6.
Reviewer 2 Report
This is a review focusing on etiology, pathogenesis, diagnosis and practical implications of hepatocellular neoplasms. Since the authors insist as practical, the relationship between treatments would be necessary. The frequently asked questions are not frequently asked practical and I don't understand the necessity of this section.
Author Response
The Guest Editor invited this review article to address important issues related to etiopathogenesis and histologic diagnosis of hepatocellular neoplasms. This field of knowledge is developing rapidly and good general reviews are available; therefore, we focused on latest developments. Treatment issues were outside the scope of our review but are covered in other invited contributions in this Cancers Special Issue (three articles have already been published and two more are expected soon).
The FAQ section was based on our experience through lecturing in international meetings and congresses, and addresses areas of new information, new methodologies and controversial issues that are of particular interest to researchers, pathologists and clinicians dealing with hepatocellular tumors. This section, too, focuses on etiopathogenesis and pathology, not treatment. We are glad that the other reviewers found this section useful.
Reviewer 3 Report
Thank you for submitting the article to the journal.
The article is very comprehensive and will benefit the readers.
I would suggest some minor upgrades to the articles.
1. Lack of tabular presentation of information makes the article hard to recapture the read content. I would suggest additions of tables such as various types of HCCs. LI-RADS table.
2. since the article mentions LI-RADS. MRI images of LI-RADS-5 HCC lesion and HCA can be added.
3. also, in the end, the FAQ section is well written. Again, a summary table of this information would be great.
4. A lesser-known entity HCC-CCA has also been mentioned which is a real added advantage of the article. However, I felt I had been admixed with other contents. I would advise making it a separate paragraph and explaining all the mentioned content, just like the authors have done for HCA.
Author Response
We thank Reviewer 3 for the kind comments.
1 and 2. We have added two tables: i) Table 1 comparing the etiology, pathogenesis, and diagnosis of HCA and HCC (also recommended by Reviewer 1); ii) Table 2 with important information related to the various subtypes of HCC. We have not added a table on LI-RADS because we have been informed by the Guest Editor that LI-RADS tables and images will be covered extensively in another invited review of this Cancers Special Issue focusing on Radiology.
- We are glad for the Reviewer’s positive comment regarding our FAQ section. However, it is difficult for us to conceive how to summarize these 16 FAQs in a table.
- We thank Reviewer 3 for the positive comment regarding combined HCC-CCA. In order to emphasize this new information, we included a separate section for combined HCC-CCA (FAQ14). We did not include this section earlier in the text because the title of the article refers to “hepatocellular neoplasms”, i.e., HCA and HCC.
Reviewer 4 Report
Article entitled “Etiology, pathogenesis, diagnosis and practical implications of hepatocellular neoplasms” by Hytiroglou P et al, highlighting the diagnostic markers of HCC is an extensive review on the topic. Authors might want to concise this lengthy article in order to help the readers. Discrepancies in the line 266 page 7 and 279 page-8 may be fixed.
Author Response
We could not think of a way of shortening the article without sacrificing content. However, we have improved the presentation of our article by adding two tables, as recommended by Reviewer 1 and Reviewer 3.
We have checked the mentioned sentences for discrepancies but could not find any.
Round 2
Reviewer 2 Report
The authors have revised the manuscript appropriately.